# Protein Modelling Highlighted Key Catalytic Sites Involved in Position-Specific Glycosylation of Isoflavonoids

**DOI:** 10.3390/ijms241512356

**Published:** 2023-08-02

**Authors:** Moon Sajid, Parwinder Kaur

**Affiliations:** UWA School of Agriculture and Environment, The University of Western Australia, 35-Stirling Highway, Perth, WA 6009, Australia; moon.sajid@research.uwa.edu.au

**Keywords:** glycosylation, UGTs, substrate and OH site specificity, isoflavonoids

## Abstract

Uridine diphosphate glycosyltransferases (UGTs) are known for promiscuity towards sugar acceptors, a valuable characteristic for host plants but not desirable for heterologous biosynthesis. UGTs characterized for the O-glycosylation of isoflavonoids have shown a variable efficiency, substrate preference, and OH site specificity. Thus, 22 UGTs with reported isoflavonoid O-glycosylation activity were analyzed and ranked for OH site specificity and catalysis efficiency. Multiple-sequence alignment (MSA) showed a 33.2% pairwise identity and 4.5% identical sites among selected UGTs. MSA and phylogenetic analysis highlighted a comparatively higher amino acid substitution rate in the N-terminal domain that likely led to a higher specificity for isoflavonoids. Based on the docking score, OH site specificity, and physical and chemical features of active sites, selected UGTs were divided into three groups. A significantly high pairwise identity (67.4%) and identical sites (31.7%) were seen for group 1 UGTs. The structural and chemical composition of active sites highlighted key amino acids that likely define substrate preference, OH site specificity, and glycosylation efficiency towards selected (iso)flavonoids. In conclusion, physical and chemical parameters of active sites likely control the position-specific glycosylation of isoflavonoids. The present study will help the heterologous biosynthesis of glycosylated isoflavonoids and protein engineering efforts to improve the substrate and site specificity of UGTs.

## 1. Introduction

Uridine diphosphate glycosyltransferases (UGTs) (E.C. 2.4.1.x) are a central player in the biosynthesis of glycosylated plant natural products (PNPs) [1]. UGTs catalyze regio- and stereo-specific glycosidic bond formation with the transfer of nucleotide-diphosphate-activated sugar moieties to the substrates and usually perform the final step in a PNP biosynthetic pathway [2,3]. UGTs involved in the biosynthesis of glycosylated PNP are universally classified into the GT1 family in the Carbohydrate Active Enzyme database (CAZy, http://www.cazy.org/) [4]. The GT1 family currently has 30,000 GTs, and the numbers are rising, with new genomes being sequenced every day [4]. However, only 1% of the GTs have been functionally characterized, which impacts the classification of glycosylated PNP biosynthetic pathways and validation of enzymatic mechanisms [5].

Over 2400 isoflavonoids have been identified, mainly from legumes [6]. Isoflavonoids’ backbone (C6-C3-C6) is biogenetically derived from flavonoids with the help of isoflavonoid synthase aided by cytochrome P450 reductase. The flavonoid backbone is formed from p-coumaroyl CoA and naringenin chalcone with the help of chalcone synthase and chalcone isomerase, respectively (Figure 1) [6,7]. Most of these isoflavonoids are found glycosylated with different sugar moieties attached to their aglycones, increasing the complexity and diversity around the C6-C3-C6 backbone [8]. Isoflavonoids generally possess a poor bioavailability in vivo due to their low solubility and poor oral absorption. However, glycosylation is a key modification that positively affects solubility, bioactivity, and bioavailability and improves the pharmacokinetic profile of isoflavonoids [9]. Additionally, isoflavonoid drugs, i.e., daidzein 8-C-glucoside currently in clinical applications, are also in the form of glycosides [10,11]. Glycosylated isoflavonoids are attracting interest due to their biochemical significance.

Recently, focus has shifted to the heterologous biosynthesis of plant natural products, and significant progress has been made in the biosynthesis of isoflavonoids [12,13,14,15,16]. The heterologous biosynthesis of isoflavonoids faces issues of a low titer and low yield. Following the development of a microbial strain capable of the de novo biosynthesis or biotransformation of an isoflavonoid of interest, efforts, i.e., directed evolution, and engineering of primary metabolism are put forward to improve the final yield [12,17]. It has been reported that the glycosylation of the end-product improves its heterologous biosynthesis rate; therefore, the addition of a UGT gene in the pathway to glycosylate the isoflavonoid of interest is a unique approach to enhance the final yield [18]. This is because isoflavonoids have a low solubility and are challenging to transport across the cell/organelles; glycosylation increases the solubility of isoflavonoids, helps in storage and transportation within and out of the cell, and drives the biosynthesis pathway forward.

Usually, plant UGTs that catalyze O-glycosylation of isoflavonoids are promiscuous and non-specific towards available OH sites (7-OH, 5-OH, and 4′-OH) [19]. Though the promiscuity of UGTs is of great value for the host plant, such plasticity negatively affects the final yield of the heterologous pathway [20,21]. Additionally, UGTs with similar amino acid sequences might have different substrate preferences, and a simple BLAST search cannot help with finding UGTs with similar substrate preferences. Therefore, the in silico characterization and ranking of available plant UGTs for the OH site and substrate specificity are required to advance the heterologous biosynthesis of O-glycosylated isoflavonoids. Thus, the present study was planned to rank selected UGTs for OH site specificity and catalysis efficiency based on multiple sequence alignment (MSA) and a protein docking analysis towards the O-glycosylation of selected isoflavonoids: genistein (GEN), biochanin-A (BCN), daidzein (DZN), formononetin (FMN), and their parent flavonoids: naringenin (NGN) and liquiritigenin (LQN). The present study will help direct the future heterologous biosynthesis of glycosylated isoflavonoids and protein engineering efforts to improve the substrate and site specificity of UGTs.

## 2. Results

### 2.1. Multiple Sequence Alignment and Phylogenetic Analysis

In total, 22 plant UGT amino acid sequences characterized for the O-glycosylation of selected (iso)flavonoids (GEN, BCN, DZN, FMN, NGN, and LQN) were retrieved from the literature (Appendix A). Multiple sequence alignment (MSA) was performed and results are given in Table 1 and Figure 2 and Appendix A. The UGT protein structure was divided into two domains: the N-terminal domain, which contains an active site that selects and holds sugar acceptors, and the C-terminal domain, which contains the plant secondary product glycosyltransferase (PSPG) motif and is responsible for binding with the sugar donor (Appendix A) [22]. Interestingly, the PSPG motif amino acid sequence of *Pueraria montana* var. lobata UGT1 and PlUGT15 were identical (Appendix A).

Only a few conserved amino acid sites (consensus threshold of ≥85%), Gly16, His17, Thr38, Asp126, Pro142, Asn222, Gly251, and Pro252 (amino acids’ number based on PlUGT15), were seen in the N-terminal domain. Due to the highest reported glycosylation efficiency and specificity towards isoflavonoids, PlUGT15 was selected as a model UGT for comparison with other UGTs [23]. The catalytic residues His17 and Asp126 were present in all UGTs except for PtUGT8, which has Asn instead of His; PlUGT57; and UGT73G1, which has Asn and Gly instead of Asp. The PSPG motif (amino acids from 348 to 391 of PlUBT15) was present in all selected UGTs and amino acids. Gln351; Glu374 that forms hydrogen bonds with the ribose ring; Asn370, Ser371, and His366 that interact with the α-phosphate group; and His366 and Ser284 that interact with the β-phosphate group of the sugar donor molecule were present in all sequences [22,24,25,26]. Additionally, Gln391 (based on the PlUGT15 sequence), which is the last amino acid residue of the PSPG motif, which plays a role in UDP-glucose recognition and specificity, was present in all selected UGT sequences, and the selected UGTs are all glucosyltransferase [22].

Phylogenetic trees were constructed using full-length and PSPG motif amino acid sequences of selected UGTs (Figure 3). The phylogenetic analysis highlighted that UGTs with the highest preference towards isoflavonoids, i.e., *Pueraria lobate* UGT15 and *Glycyrrhiza uralensis* UGT3, emerged recently. Selected UGTs have likely accumulated mutations in their coding sequence at a different rate, mainly in the N-terminal domain, compared with the C-terminal domain and the PSPG motif (Figure 2). Furthermore, the accumulation of amino acid substitutions in the N-terminal domain has likely led to a higher substrate specificity and efficiency towards isoflavonoids. However, further research is required to unfold the relationship between different substitution rates between N- and C-terminal domains, substrate specificity, and the rate of catalysis.

### 2.2. Protein Modeling and Docking Analysis

To analyze biochemical and structural parameters that define the substrate and OH site specificity towards selected isoflavonoids, protein models of selected UGTs were developed with Alphafold (Table 2, Table 3, and Appendix A) [27]. A very high per-residue confidence score (pLDDT score > 90) was seen for key catalytic residues (Asp), the PSPG motif region, and the donor binding site of all UGT models. For catalytic His, 15 models showed a very high confidence (pLDDT > 90) and 7 models showed a good confidence (pLDDT 90> and <70) in His’s predicted orientation and position (Appendix A). Based on the pLDDT score, the predicted models of selected UGTs, position and orientation of catalytic residues, PSPG motif region, and donor binding site were considered reliable [28]. Protein 3D structures of UGT71G1, *Vitis vinifera* UGT1, UGT85H2, and UGT78G1 that can glycosylate isoflavonoids were resolved and could be used for homology modelling and comparison [22,24,25,26]. However, homology models developed through Alphafold are considered the most accurate and reliable compared with other methods, even for challenging proteins [27,29]. Advances in protein structure modelling have great potential to help with the better understanding of sequence–structure relationships.

Selected UGTs were divided into three groups based on preferred OH sites (7-OH, 5-OH, and/or 4′-OH) and the active site structure (Table 1). The docking scores and distance between catalytic His and the OH group (docked closest to catalytic His) are given in Table 3. A higher docking score indicates a better binding efficiency, and the UGTs with a higher docking score and preference for 7-OH sites are mainly present in group 1. The UGTs with comparatively low docking scores and that are non-specific towards available OH sites were placed in groups 2 and 3. Orientations and interactions of selected substrates with their UGT model and UGT–UGT structural comparison have highlighted active sites’ physical and chemical parameters that define the substrate and site-specific glycosylation of isoflavonoids; however, biochemical experiments are required to verify and support these findings.

Group 1 contains seven UGTs, and their active sites are almost identical in the size and composition of amino acids. Group 1 UGTs predominantly prefer the 7-OH site for glycosylation, and they prefer selected isoflavonoids over flavonoids, as is clear from the docking score and distance between catalytic His and the 7-OH group (Figure 4 and Appendix A and Table 3). The presence of four conserved aromatic amino acids (Phe127, Tyr147, Phe157, and Phe206; amino acid numbers based on PlUGT15) likely plays a role in maintaining the 3D architecture of group 1 UGTs and subsequently facilitates 7-OH-site-specific glycosylation. The highest docking score for selected substrates was seen for *Glycyrrhiza uralensis* GT6. GuUGT6 has Tyr in place of Phe206, which develops a polar interaction with the C-ring keto group of all tested substrates and might be the reason for the highest fitness score compared with other models. However, the information about the catalytic efficiency and specific activity of GuGT6 towards GEN, BCN, LQN, and NGN has not been reported, therefore PlUGT15 was kept as a model UGT due to its reported catalytic efficiency and activity towards all selected substrates except BCN [23,30].

Group 1 UGTs also have a string of 19 conserved amino acids in their PSPG motif (from His357 to Ala375; amino acid numbers based on PlUGT15). These amino acids are also likely to play a role in the maintenance of the active site 3D structure and subsequently play a role in substrate orientation and site specificity. Tyr147 and Ser149 (amino acid numbers based on PlUGT15) are present in all group 1 UGTs, and these amino acids interact with either the C-ring keto or ether group of substrates analyzed in the present study. Ser149 also develops an additional interaction with the 5-OH of BCN, GEN, and NGN. As C-ring keto and ether groups are present in all the selected substrates, these amino acids are unlikely to play a role in substrate selection but may facilitate 7-OH-specific glycosylation. Based on these findings, it is likely that the conserved and compact structure of the active site plays a role in substrate selection, and the polar interactions probably play a role in OH site specificity. Therefore, a UGT with a structure like a group 1 UGT will possibly catalyze the 7-OH-site-specific glycosylation of isoflavonoids at a comparatively higher catalytic efficiency than other PNPs.

Group 2 contains eight UGTs, and for a group-specific analysis and comparison, UGT72V3 was selected as the model UGT due to its highest docking score, and other UGTs were compared with it. Overall, the active sites of group 2 UGTs are comparatively broader and more diverse in overall 3D architecture and amino acid composition (Figure 4 and Appendix A). Therefore, UGTs in group 2 can accept different substrates/allow different orientations and subsequently glycosylate any of the available OH sites (7-OH, 5-OH, and 4′-OH) at an almost equal rate as shown with the fitness score and preferred OH sites (Table 3). Except for Phe132 (amino acid numbers based on UGT72V3), no other conserved aromatic amino acid of group 1 UGTs is found in group 2. The PSPG motif of group 2 UGTs only has Ala388-Glu389-Gln390 (amino acid numbers based on UGT72V3) in place of the 19 conserved amino acid string of group 1. The interaction between the C-ring keto or ether group of substrates and active site amino acids in the top-ranking ligand orientation of group 2 UGTs was completely missing. The only significant interaction, other than catalytic His, was between the 7-OH group of substrates and active site amino acids of group 2 UGTs. The group 2 UGT active site, being wide and open, provides enough space for substrates to orient, with either the 7-OH or 5-OH site docked, close to catalytic His, and thus can glycosylate any of these OH groups at an equal rate. These active site features allow UGTs to accept different substrates/OH sites for glycosylation but likely compromise the catalysis rate.

Group 3 has five UGTs, and due to the highest docking score, UGT73F1 was selected as the model UGT for a group-specific analysis and comparison. Overall, the active sites of group 3 UGTs are chemically and structurally diverse and wider than group 1 and group 2 UGTs (Figure 4 and Appendix A). Group 3 UGTs can accept different substrates and prefer the 7-OH site but can glycosylate 5-OH and 4′-OH sites, and this might be the reason for the low fitness score and higher distance between catalytic His and the closest docked OH group. Only one aromatic amino acid, Phe/Met at position 19 (amino acid numbers based on UGT74F1), is seen in the active sites of group 3 UGTs. Phe/Met develops a polar interaction with the C-ring keto group of all substrates and this polar interaction with the C-ring keto group seems to be the only reason for the preference for 7-OH site-specific glycosylation (Table 3). His381 is another conserved active site amino acid (amino acid number based on UGT73F1); however, His381 is likely not playing any part in the orientation of substrates due to the distance from docked substrates (>4 Å). No other interaction between the substrate and active site amino acids was seen for group 3 UGTs except with catalytic His. So, the wider active site enables group 3 UGTs to accept a range of substrates, mostly flavonoids.

*Glycine max* UGT4 and PlUGT57 have the lowest fitness score and have very different orientations of substrates and active site structures (Appendix A). The comparison of GmUGT4 and PlUGT57 with the model UGT highlighted a different arrangement of loops around the active site, which might be the reason for the low fitness scores (Appendix A). Additionally, it is reported that PlUGT57 does not accept NGN and LQN. Based on the docking analysis and active site structures, we argue that PlUGT57 could glycosylate NGN and LQN. However, the rate of catalysis might be very low, and the amount formed during the chemical reaction might be challenging to detect. It is also possible that these UGTs might use a different mechanism of catalysis/catalytic amino acids that can be analyzed with a different docking analysis pipeline (see Appendix B).

## 3. Discussion

In the present study, UGTs that can glycosylate (iso)flavonoids (GEN, BCN, DZN, FMN, NGN, and LQN) are analyzed for OH site specificity and the catalysis rate. Certain structural and chemical features of active sites that likely define the substrate and OH site specificity are documented. The MSA and phylogenetic analysis show a low sequence similarity among selected UGTs, highlighting a higher conservation of PSPG motif and C-terminal domain amino acids than the N-terminal domain. The conserved active site amino acids identified in this study are likely to play a structural and chemical role in isoflavonoid-specific 7-OH glycosylation. The functional characterization of proposed critical amino acid sites will help design a UGT with the highest preference and efficiency towards a substrate and OH site [41]. Site-directed mutagenesis, directed evolution, and rotational protein design can help engineer UGTs with the required characteristics [4,42]. Interestingly, most of the selected UGTs with the highest substrate preference for isoflavonoids and 7-OH site specificity are expressed in the chloroplast. Such compartmentalization may help plants store isoflavonoids following glycosylation and is likely the reason for the higher concentration of isoflavonoids compared with heterologous biosynthesis [43,44].

The role of UGTs in detoxifying xenobiotics (allelochemicals before the introduction of synthetic chemicals) likely has driven the evolution of new UGTs with different amino acid compositions around the acceptor-binding active site [25]. Isoflavonoid biosynthesis is a recently evolved biochemical pathway linked with the emergence of isoflavonoid synthase (IFS) [6,7]. It is likely that UGTs specific for the glycosylation of isoflavonoids have co-evolved with IFS and have gained specificity for said substrates over time [45,46]. Our results highlight that amino acid substitutions accumulated in the N-terminal domain during evolution have helped UGTs improve the substrate preference, OH site specificity, and glycosylation efficiency for isoflavonoids. Most of the active site amino acids of group 1 UGTs are present in the helices compared with groups 2 and 3, which have active site amino acids in loops, which further support structural evolution and the physical role of the active site in the enzyme’s specificity and catalysis efficiency.

Usually, the promiscuity of UGTs is considered advantageous for plants [4]. However, from an enzyme’s specificity and heterologous biosynthesis point of view, UGT promiscuity raises two critical questions: (i) if UGTs are promiscuous, why does the plant genome encode multiple UGT genes, and (ii) how is the expression of UGTs regulated in the plant cell to produce only a limited number of products at a specific time? The latter question can be answered based on the genetic mechanism that regulates the biosynthesis of a compound of interest. So, if the biosynthesis of the compound of interest is off, no glycosylated product of that pathway will be formed, even if the UGTs are being expressed. However, multiple UGT genes per genome could be linked with the diversity of PNPs synthesized in plants [4,11,47]. Instead of one UGT–one PNP, UGT promiscuity lowers the burden from UGT regulatory pathways and helps biochemical-pathway-specific genetic regulation to be more effective. However, further studies are required to understand the reasons for UGTs’ promiscuity and gene expression regulation [48]. Such studies will also help heterologous isoflavonoid biosynthesis by providing data about the biosynthesis of a specific substrate during a specific time at an industrially acceptable titer and yield.

## 4. Materials and Methods

### 4.1. UGT Sequence Selection and Retrieval

UGTs with reported glycosylation activity towards selected isoflavonoids (daidzein, formononetin, genistein, and biochanin-A) and flavonoids (naringenin and liquiritigenin) were searched in the literature using keywords “isoflavonoids” and “glycosylation” in Google Scholar and PubMed. The amino acid sequences were retrieved from NCBI and subjected to a downstream analysis (Appendix A).

### 4.2. Phylogenetic Analysis and Protein Modelling

The UGT amino acid sequences were aligned using the MUSCLE alignment algorithm with default parameters in Geneious Prime (version 2021.1.1). Following this, a phylogenetic tree was constructed using the neighbor-joining algorithm with default parameters on Geneious Prime (version 2021.1.1), and the results were documented. Protein 3D models of selected UGT amino acid sequences were developed using the Alphafold server (https://alphafold.ebi.ac.uk/ (accessed on 11 May 2023)) (Appendix A). The physical and chemical parameters of selected UGTs were analyzed using ProtParam online tool (http://www.expasy.org/tools/protparam.html (accessed on 11 June 2023)). The subcellular localizations of selected UGTs were determined with WoLF PSORT (https://wolfpsort.hgc.jp/ (accessed on 11 June 2023)).

### 4.3. Protein Docking Analysis

For a protein docking analysis, the automated docking program GOLD (version 2022.3.0) was used [49]. Substrates (GEN, BCN, DZN, FMN, NGN, and LQN) were docked with selected UGTs, and the docking was restricted to the binding pocket by defining active site amino acids (Appendix A). All docking parameters were kept as default. The CHEMPLP fitness scoring function was used to identify the lowest energy docking results. (Please see Appendix B for details of the docking analysis that includes a comparison of different active site/docking pocket amino acid combinations and comparison of CHEMPLP vs. the GOLD-Score used to rank the best ligand orientation). The docking results were visualized and analyzed using PyMOL (version 4.6.0). For the comparison of the active site 3D structure and chemical composition, amino acids around the substrates within 5 Å were analyzed.

## 5. Conclusions

UGTs with reported glycosylation activity towards an isoflavonoid showed a low amino acid sequence similarity. N- and C-terminal domains of selected UGTs accumulated amino acid substitutions at different rates. It is likely that the N-terminal domain amino acid sequence evolved faster to gain specificity for the glycosylation of an isoflavonoid. Based on the docking score, OH site specificity, active site structure, and amino acid composition, selected UGTs were placed in three groups. UGTs, i.e., GuGT6 and PlUGT15, with the highest docking score and 7-OH site-specific glycosylation, were grouped in group 1. A significantly higher pairwise identity (67.4%) and identical sites (31.7%) were seen for group 1 UGTs compared with other groups. The protein modelling and docking analysis showed that active sites’ structure and chemical composition defined the substrate preference and OH site specificity. Protein engineering efforts can help design and improve the substrate and site specificity of UGTs towards a substrate or PNP family.

## Figures and Tables

**Figure 1 ijms-24-12356-f001:**
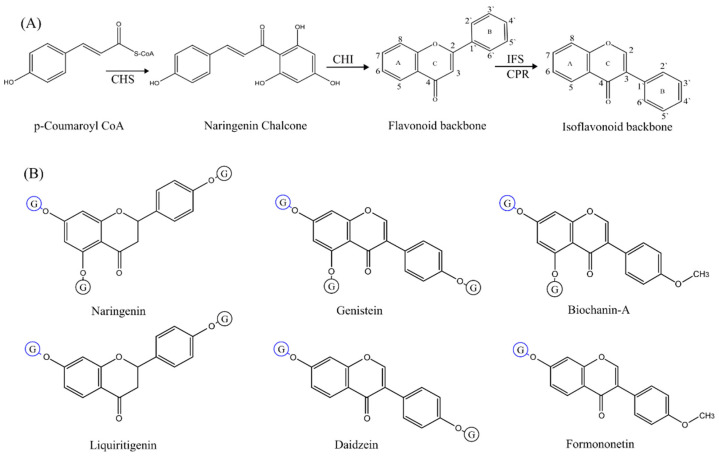
Biosynthetic pathway of isoflavonoids and chemical structures of selected substrates. (**A**) Chalcone synthase (CHS) converts p-coumaroyl Co-A into naringenin chalcone, which is isomerized by chalcone isomerase (CHI) to produce the flavonoid structure (C6-C3-C6 backbone). Isoflavonoid synthase (IFS) and cytochrome P450 reductase (CPR) perform an aryl ring migration reaction (B-ring) to produce an isoflavonoid backbone from flavonoids. (**B**) Chemical structures of naringenin (NGN), genistein (GEN), biochanin-A (BCN), liquiritigenin (LQN), daidzein (DZN), and formononetin (FMN) are given. Glycosylation sites have been shown with the letter ‘G’ inside a blue (most common glycosylation sites) or a black circle (less common glycosylation sites).

**Figure 2 ijms-24-12356-f002:**
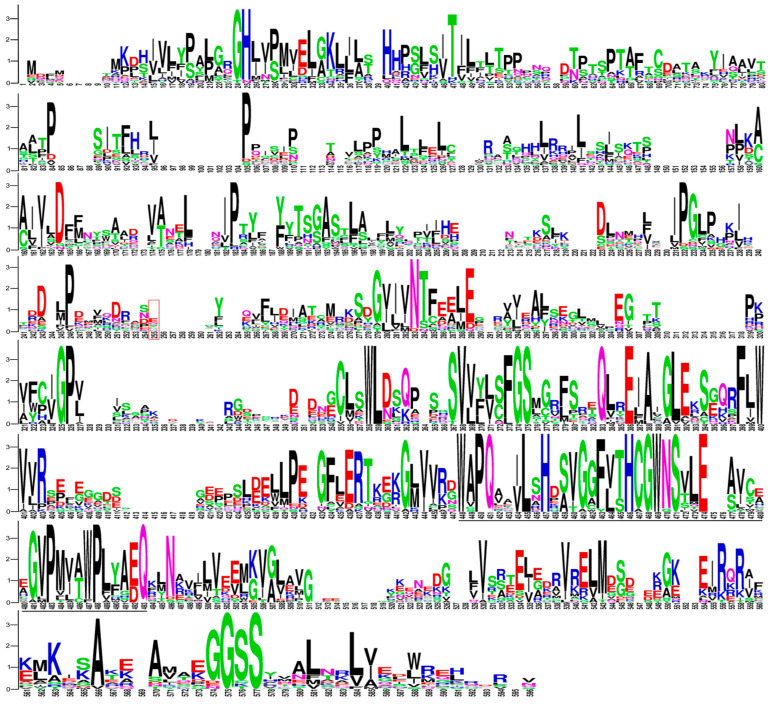
Sequence logos of the multiple sequence alignments of the 22 UGTs’ amino acid sequences. The sequence logos are based on UGTs’ amino acid alignment using the neighbor-joining algorithm on Geneious Prime. The logos were generated using Weblogo (https://weblogo.berkeley.edu/logo.cgi (accessed on 11 July 2023)). The bit score indicates the information content for each position in the sequence. Amino acids are shown with single-letter symbols, and the height of the letter at each position represents the degree of conservation. The plant secondary product glycosyltransferase (PSPG) motif is underlined with a black line. The red box represents amino acids up to the N-terminal domain.

**Figure 3 ijms-24-12356-f003:**
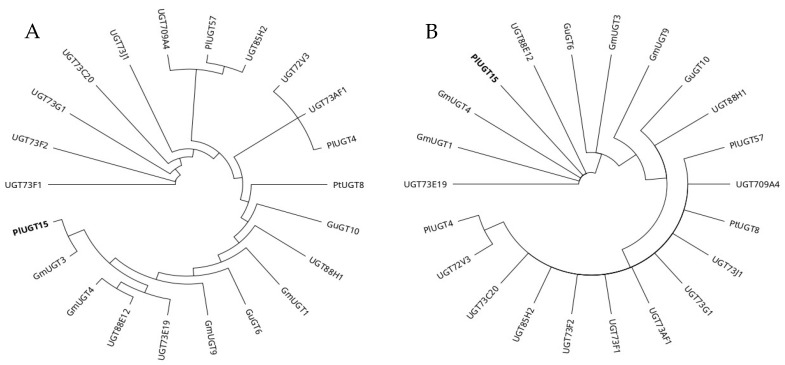
Phylogenetic tree of (**A**) full-length and (**B**) PSPG motif amino acid sequences of selected UGTs. The phylogenetic tree was constructed using the neighbor-joining method with default parameters in Geneious prime version 2022.1.1. Due to the highest reported glycosylation efficiency and specificity towards isoflavonoids, PlUGT15 (bold) was used as a model UGT for comparison with other UGTs.

**Figure 4 ijms-24-12356-f004:**
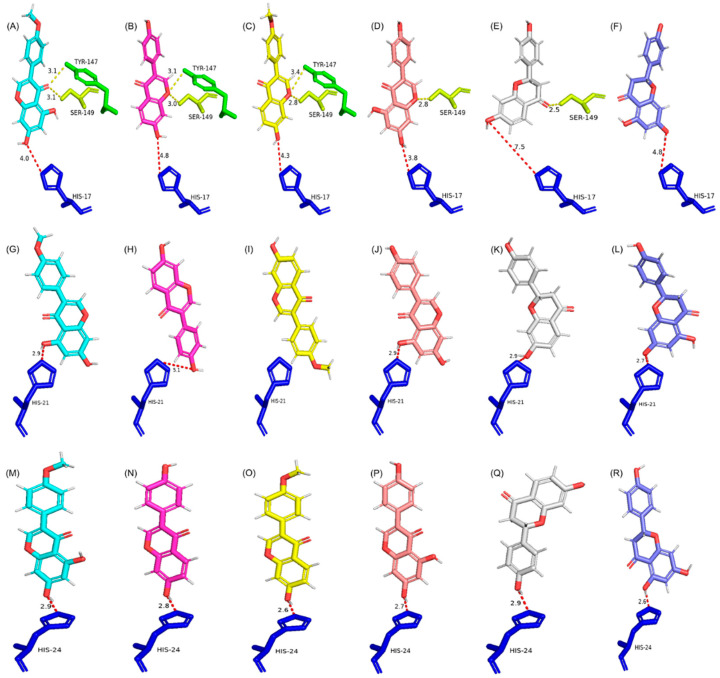
Docking results of PlUGT15 (**A**–**F**), UGT72V3 (**G**–**L**), and UGT73F1 (**M**–**R**) with selected substrates. Different orientations of catalytic His in selected models highlight distinct active site structures that result in the different orientation of substrates (the substrates are shown in vertical orientations for better visuals and comparison). Yellow dotted lines represent polar interactions between substrates and amino acids; red dotted lines represent the distance between catalytic His and the closest OH group. The distance is given in Å units. Key amino acids and substrates are shown in stick models. Catalytic His is colored blue, Tyr in green, and Ser in yellow. The carbon skeleton of biochanin-A, daidzein, formononetin, genistein, liquiritigenin, and naringenin is colored cyan, pink, yellow, salmon, grey, and slate, respectively, and the oxygen atoms are colored red.

**Table 1 ijms-24-12356-t001:** Percentage of pairwise identity and identical sites of selected UGTs.

Description	Total UGTs	Pairwise Identity (%)	Identical Site (%)	Preferred OH Sites
All UGTs	22	33.2	4.5	n.a.
N-terminus	22	25.1	0.3	n.a.
C-terminus	22	44.1	9.1	n.a.
PSPG motif	22	67.9	28.3	n.a.
Group 1	7	67.4	31.7	7-OH
Group 2	8	28.6	8.9	All 3 OH
Group 3	5	28.3	8.2	7 and 4′ OH

n.a.: Not applicable.

**Table 2 ijms-24-12356-t002:** Biochemical characteristics of selected UGTs.

Group	Name	Amino Acids	RMSD ^(a)^	Mol. Weight	Isoelectric Point	Aliphatic Index	Instability Index	Subcellular Location	Ref.
G1	GuGT6	486	0.51	53.40	6.24	93.83	Unstable	Cytoplasm	[30]
	PlUGT15	472	Model	52.34	5.87	90.64	Stable	Chloroplast	[23]
	UGT73E19	473	1.46	52.75	5.60	90.47	Stable	Chloroplast	[31]
	GmUGT9	468	0.58	51.50	6.76	100.13	Unstable	Chloroplast	[32]
	GmUGT3	473	0.30	52.29	5.51	91.52	Stable	Chloroplast	[32]
	UGT88E12	471	0.62	52.20	5.48	92.72	Unstable	Chloroplast	[33]
	UGT88H1	451	1.61	50.26	8.10	87.49	Unstable	Chloroplast	[33]
G2	UGT72V3	474	1.95	52.37	5.14	97.64	Unstable	Chloroplast	[31]
	UGT72AF1	469	1.30	51.95	6.12	99.74	Unstable	Chloroplast	[31]
	GuGT10	462	1.06	51.53	5.99	94.48	Unstable	Chloroplast	[30]
	UGT73J1	469	1.75	52.95	5.36	83.94	Stable	Chloroplast	[34]
	UGT73G1	479	1.42	53.58	6.08	93.01	Unstable	Chloroplast	[34]
	GmUGT1	474	0.86	52.04	4.96	95.08	Unstable	Chloroplast	[35]
	PlUGT4	465	1.61	51.08	5.85	102.90	Unstable	Cytoplasm	[23]
	PtUGT8	426	1.81	47.71	5.86	99.74	Unstable	Cytoplasm	[36]
G3	UGT73F1	482	1.73	54.32	5.64	86.37	Unstable	Cytoplasm	[37]
	UGT73F2	476	1.22	53.24	6.46	84.20	Unstable	Chloroplast	[38]
	UGT709A4	480	1.21	50.87	6.29	94.79	Unstable	Chloroplast	[39]
	UGT85H2	482	0.78	54.50	5.33	88.13	Stable	Chloroplast	[22,40]
	UGT73C20	502	0.32	56.82	6.95	91.24	Stable	Chloroplast	[31]
	GmUGT4	473	0.62	52.25	5.70	92.54	Stable	Chloroplast	[32]
	PlUGT57	477	0.90	52.72	6.24	88.47	Stable	Cytoplasm	[23]

^(a)^ For root mean square deviation (RMSD), protein models were compared with PlUGT15.

**Table 3 ijms-24-12356-t003:** UGT grouping based on docking score and OH site specificity.

Group	UGTs	GEN			BCN			DZN			FMN			LQN			NGN		
		Fitness Score	OH Site	Dis. to His ^(a)^	Fitness Score	OH Site	Dis. to His	Fitness Score	OH Site	Dis. to His	Fitness Score	OH Site	Dis. to His	Fitness Score	OH Site	Dis. to His	Fitness Score	OH Site	Dis. to His
G1	GuGT6	74.90	7	3.8	80.80	7	3.9	73.40	7	3.9	79.45	7	3.9	64.79	7	6.6	71.52	7	5.6
	PlUGT15	70.00	7	3.8	76.14	7	4.0	70.81	7	4.8	75.59	7	4.3	66.66	7	7.5	66.93	5	4.8
	UGT73E19	74.76	7	4.3	73.80	7	3.8	73.91	7	4.3	71.33	7	3.6	66.45	7	6.8	7.40	7	5.6
	GmUGT9	66.96	7	4.2	71.91	(b)	-	66.33	7	4.2	72.67	7	4.0	67.84	7	3.1	61.90	5	2.9
	GmUGT3	67.18	7	4.4	69.39	7	3.9	65.69	7	4.0	67.79	7	3.9	57.57	4′	3.0	60.49	4′	2.8
	UGT88E12	53.93	7	3.1	52.91	(c)	-	57.29	4′	2.8	58.75	7	3.1	49.20	-	-	51.52	5	2.9
	UGT88H1	66.00	7	3.1	67.94	7	3.0	63.06	4′	2.7	61.90	7	3.0	63.43	7	2.9	61.19	7	2.8
G2	UGT72V3	58.36	5	2.9	60.58	5	2.9	56.97	4′	5.1	61.24	(b)	-	54.44	7	2.9	52.42	7	H1.8
	UGT72AF1	55.89	4′	3.6	56.84	7	2.1	55.47	4′	3.6	56.67	7	2.9	60.62	4′	3.1	58.24	4′	3.0
	GuGT10	53.11	(c)	-	54.82	(c)	-	56.14	-	-	57.82	(b)	-	54.90	-	-	57.52	5	3.0
	UGT73J1	52.73	(c)	-	54.37	(c)	-	54.30	4′	2.8	55.41	(b)	-	56.96	4′	2.6	55.10	5	2.8
	UGT73G1	51.35	5	3.1	54.36	5	3.0	48.59	7	5.1	50.17	7	5.6	51.22	7	2.8	51.89	5	4.7
	GmUGT1	58.89	4′	2.9	63.98	7	3.0	61.56	4′	3.0	64.84	7	2.7	60.58	4′	4.8	59.59	5	3.0
	PlUGT4	54.34	7	2.6	50.35	7	2.9	54.91	7	2.6	52.74	7	2.5	55.22	4′	2.5	53.66	4′	2.9
	PtUGT8	50.10	7	2.6	50.19	7	2.6	50.75	7	2.9	52.79	7	2.7	53.02	7	2.9	55.24	5	2.5
G3	UGT73F1	47.13	7	2.7	50.30	7	2.9	47.25	7	2.8	50.39	7	2.6	46.00	4′	2.9	44.48	5	2.6
	UGT73C20	36.10	4′	4.4	38.81	5	4.3	32.31	7	4.4	40.60	(b)	-	38.31	4′	3.6	38.20	4′	4.3
	UGT73F2	47.78	4′	3.0	48.46	(c)	-	47.05	4′	3.0	47.70	7	3.0	48.50	7	2.9	45.26	4′	3.2
	UGT709A4	40.99	7	4.2	33.10	7	4.3	42.41	7	4.2	34.63	7	4.2	47.47	-	-	46.77	-	-
	UGT85H2	51.52	7	5.9	42.08	(c)	-	52.41	7	6.0	49.39	7	7.2	54.69	7	5.7	49.46	7	6.0

^(a)^ The distance from the nearest OH group of substrates to the τN atom of catalytic His was calculated. (b) The Methyl group docked towards catalytic His. (c) The available OH groups docked at almost equal distances from catalytic His.

## Data Availability

Not applicable.

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
