# Peer review of "Protein Modelling Highlighted Key Catalytic Sites Involved in Position-Specific Glycosylation of Isoflavonoids"

_ijms, 2023, doi:10.3390/ijms241512356_

Round 1
Reviewer 1 Report
Comments and Suggestions for Authors
This manuscript mainly uses MSA and docking to study the sequence and structural basis of substrate preference for 22 proteins in the UGT family. As for the content of the current manuscript, there are still many places worth discussing.
1. The critical points expressed in the text should be highlighted in the main text. Attention should be paid to the use of figures and tables.
a. The results of the MSA in the results section should be displayed directly in the figures and tables of the main text.
b. Lines 104-106 of the main text 'Selected UGTs have likely accumulated mutations at a different rate in their coding sequence, mainly in the N-terminal domain, compared with the C-terminal domain and the PSPG motif' are not specifically displayed.
c. The reason for choosing PlUGT15 as the comparison template should be explained in the main text, not the figure caption.
d. The classification criteria for the three groups of UGTs are not clear enough.
e. It is recommended to display the docking results of all enzyme substrates.
2. The reason for choosing PlUGT15 as a comparison template is 'highest reported glycosylation efficiency and specificity towards isoflavonoids' (lines 116-117) is justified. However, using the structures published in PDB (such as UGT85H2, etc.) as a comparison template is still a good supplement.
In addition, why does PlUGT15 have the highest glycosylation efficiency and specificity towards isoflavonoids but not the highest docking score (lines 135-136)? Does this indicate a flaw in the research method?
3. For the use of the prediction model, how to determine that the alphafold2 prediction model is credible, especially for the key motifs?
4. The differently stained structures in Figure 2 are not clearly explained. And what interactions does the yellow line show? It is recommended to display the key points of all enzymes involved in the paper in the supplementary figures.
5. The format of the Figure S2 caption needs to be unified with the previous ones.
6. What is the alignment standard for the models in Figure S3?
7. Biochemical experiments to verify the impact of key sites found by MSA and molecular docking on enzyme activity are crucial and necessary.
Comments on the Quality of English LanguageThe logic of the manuscript is not rigorous enough and sometimes not smooth to read.
a. The PSPG motif was first introduced in line 79 of the main text, but the introduction of the UGT structural domain is in line 85. And it is necessary to visually display the structural motifs of the N-terminus, C-terminus, etc.
b. The 'However' in line 49 of the main text doesn't seem to be quite appropriate logically. The progress made in the synthesis of glycosylated isoflavones in the previous sentence and the low drop rate problem of isoflavone synthesis in this sentence, as well as the contradiction in logic that glycosylation has increased the drop rate of isoflavones.
c. The introduction from lines 67-72 should include a summary of the research methods used later.
Author Response
Reviewer 1:
Comments and Suggestions for Authors
This manuscript mainly uses MSA and docking to study the sequence and structural basis of substrate preference for 22 proteins in the UGT family. As for the content of the current manuscript, there are still many places worth discussing.
- The critical points expressed in the text should be highlighted in the main text. Attention should be paid to the use of figures and tables.
Response: Two new figures (Figure 1. Biosynthetic pathway of isoflavonoids and chemical structures of selected substrates & Figure 2. Sequence logos of the multiple sequence alignments of the 22 UGTs amino acids sequences) and a new table (Table 1. Percentage of pairwise identity and identical sites of selected UGTs) have been added to the main text to highlight key findings. Additionally, figure 4 (Figure 4: Docking results of PlUGT15 (A to F), UGT72V3 (G to L) and UGT73F1 (M to R) with selected substrates) has been improved to highlight the key findings of docking analysis.
Captions of figures and tables added/improved during the revision process:
Figure 1. Biosynthetic pathway of isoflavonoids and chemical structures of selected substrates. (A) Chalcone synthase (CHS) converts p-coumaroyl Co-A into naringenin chalcone which is isomerized by chalcone isomerase (CHI) to give flavonoid structure (C6-C3-C6 backbone). Isoflavonoid synthase (IFS) and cytochrome P450 reductase (CPR) perform an aryl ring migration reaction (B-ring) to produce an isoflavonoid backbone from flavonoids. (B) Chemical structures of naringenin (NGN), genistein (GEN), biochanin-A (BCN), liquiritigenin (LQN), daidzein (DZN) and formononetin (FMN) are given. Glycosylation sites have been shown with the letter ‘G’ inside a blue (most common glycosylation sites) or a black circle (less common glycosylation sites).
Figure 2. Sequence logos of the multiple sequence alignments of the 22 UGTs amino acids sequences. The sequence logos are based on UGTs amino acids alignment using the neighbour-Joining algorithm on Geneious Prime. The logos were generated using Weblogo (https://weblogo.berkeley.edu/logo.cgi (accessed on 11 July 20223)). The bit score indicates the information content for each position in the sequence. Hight of the letters designating the amino acid residues at each position represents the degree of conservation. The plant secondary product glycosyltransferase (PSPG) motif is underlined with a black line. The red box represents amino acids up to the N-terminal domain.
Table 1. Percentage of pairwise identity and identical sites of selected UGTs
Figure 4. Docking results of PlUGT15 (A to F), UGT72V3 (G to L) and UGT73F1 (M to R) with selected substrates. Different orientations of catalytic His in selected models highlights distinct active site structures that result in the different orientation of substrates (the substrates are shown in vertical orientations for better visuals and comparison). Yellow dotted lines represent polar interactions between substrates and amino acids, red dotted lines represent the distance between catalytic His and the closest OH group. The distance is given in â„« units. Key amino acids and substrates are shown in stick models. Catalytic His is coloured blue, Tyr in green and Ser in yellow. The carbon skeleton of biochanin-A, daidzein, formononetin; genistein, liquiritigenin and naringenin is coloured cyan, pink, yellow, salmon, grey, and slate respectively, and the oxygen atoms are coloured red.
- The results of the MSA in the results section should be displayed directly in the figures and tables of the main text.
Response: The results of MSA (all UGTs, N & C terminal domains, PSPG motif, and group 1-3) have been moved from the text to a new table (Table 1. Percentage of pairwise identity and identical sites of selected UGTs). And a new figure (Figure 2. Sequence logos of the multiple sequence alignments of the 22 UGTs amino acids sequences) highlighting MSA results as amino acid logos has been added to the main text.
Captions of figures and tables added during the revision process:
Table 1. Percentage of pairwise identity and identical sites of selected UGTs
Figure 2. Sequence logos of the multiple sequence alignments of the 22 UGTs amino acids sequences. The sequence logos are based on UGTs amino acids alignment using the neighbour-Joining algorithm on Geneious Prime. The logos were generated using Weblogo (https://weblogo.berkeley.edu/logo.cgi (accessed on 11 July 2023)). The bit score indicates the information content for each position in the sequence. Hight of the letters designating the amino acid residues at each position represents the degree of conservation. The plant secondary product glycosyltransferase (PSPG) motif is underlined with a black line. The red box represents amino acids up to the N-terminal domain.
- Lines 104-106 of the main text 'Selected UGTs have likely accumulated mutations at a different rate in their coding sequence, mainly in the N-terminal domain, compared with the C-terminal domain and the PSPG motif' are not specifically displayed.
Response: A new figure (Figure 2. Sequence logos of the multiple sequence alignments of the 22 UGTs amino acids sequences) has been developed to display MSA and the rate of mutations in the N-and C-terminal domain and the PSPG motif sequence. The MSA results are developed as amino acid logos. The logos were generated using Weblogo and the bit score indicates the information content for each position in the sequence. Hight of the letters designating the amino acid residues at each position represents the degree of conservation.
Captions of Figure 2 developed during the revision process:
Figure 2. Sequence logos of the multiple sequence alignments of the 22 UGTs amino acids sequences. The sequence logos are based on UGTs amino acids alignment using the neighbour-Joining algorithm on Geneious Prime. The logos were generated using Weblogo (https://weblogo.berkeley.edu/logo.cgi (accessed on 11 July 20223)). The bit score indicates the information content for each position in the sequence. Hight of the letters designating the amino acid residues at each position represents the degree of conservation. The plant secondary product glycosyltransferase (PSPG) motif is underlined with a black line. The red box represents amino acids up to the N-terminal domain.
- The reason for choosing PlUGT15 as the comparison template should be explained in the main text, not the figure caption.
Response: The reason for choosing PlUGT15 as model UGT has been added in the main text (Lines 103 to 105).
Text from lines 103 to 105 is given below:
Due to the highest reported glycosylation efficiency and specificity towards isoflavonoids, PlUGT15 is selected as a model UGT for comparison with other UGTs [23].
- The classification criteria for the three groups of UGTs are not clear enough.
Response: The clarity of classification criteria for three groups of UGTs has been improved (Lines 149-150).
Text from lines 149 to 150 is given below:
Selected UGTs were divided into three groups based on preferred OH sites (7-OH, 5-OH, and/or 4’-OH) and active site structure (Table 1).
- It is recommended to display the docking results of all enzyme substrates.
Response: Figure 4 (figure number is changed due to the addition of new figures) has been improved to show the docking results of all substrates for selected UGTs. Additionally, figures S4A to S4C have been developed to show the docking results of all UGTs with selected substrates.
Captions of Figure 4 improved during the revision process:
Figure 4. Docking results of PlUGT15 (A to F), UGT72V3 (G to L) and UGT73F1 (M to R) with selected substrates. Different orientations of catalytic His in selected models highlights distinct active site structures that result in the different orientation of substrates (the substrates are shown in vertical orientations for better visuals and comparison). Yellow dotted lines represent polar interactions between substrates and amino acids, red dotted lines represent the distance between catalytic His and the closest OH group. The distance is given in â„« units. Key amino acids and substrates are shown in stick models. Catalytic His is coloured blue, Tyr in green and Ser in yellow. The carbon skeleton of biochanin-A, daidzein, formononetin; genistein, liquiritigenin and naringenin is coloured cyan, pink, yellow, salmon, grey, and slate respectively, and the oxygen atoms are coloured red.
- The reason for choosing PlUGT15 as a comparison template is 'highest reported glycosylation efficiency and specificity towards isoflavonoids' (lines 116-117) is justified. However, using the structures published in PDB (such as UGT85H2, etc.) as a comparison template is still a good supplement.
In addition, why does PlUGT15 have the highest glycosylation efficiency and specificity towards isoflavonoids but not the highest docking score (lines 135-136)? Does this indicate a flaw in the research method?
Response: The comparison of selected UGT protein models with UGT85H2 is logical and was tested during the UGT protein models development and optimisation of the docking analysis pipeline (Appendix A). Firstly, UGT models developed with Alphafold have shown better protein model alignment scores (RMSD scores) with PlUGT15 compared with UGT85H2. Secondly, protein models of selected UGTs were also developed with the SWISS-MODEL tool using UGT85H2 as a template. However, as mentioned in lines 145-148, UGT models developed with Alphafold performed better compared with models developed with the SWISS-MODEL tool. Therefore, UGT models developed with Alphafold were used for analysis.
The reason for the highest docking score for GuGT6 has been mentioned in the text (lines 166-172), however, the information about catalytic efficiency and specific activity of GuGT6 towards GEN, BCN, LQN and NGN have not been reported, therefore PlUGT15 was kept as model UGT due to its reported catalytic efficiency and activity towards all selected substrates except BCN. And the reason to keep PlUGT15 as model UGT has been added in the text (Lines: 166-172).
Text from lines 166 to 172 is given below:
The highest docking score for selected substrates was seen for Glycyrrhiza uralensis GT6. GuUGT6 has Tyr in place of Phe206, which develops a polar interaction with the C-ring keto group of all tested substrates and might be the reason for the highest fitness score compared with other models. However, the information about catalytic efficiency and specific activity of GuGT6 towards GEN, BCN, LQN and NGN have not been reported, therefore PlUGT15 was kept as model UGT due to its reported catalytic efficiency and activity towards all selected substrates except BCN [23,29].
- For the use of the prediction model, how to determine that the alphafold2 prediction model is credible, especially for the key motifs?
Response: Alphafold2 has been reported to be the best prediction model developed so far (Lines 145-148). Additionally, we have not seen any artifacts for key motifs in UGT models developed with Alphafold2 when compared with UGT PDB models. Additionally, UGT models developed with Alphafold have shown better protein model alignment scores (RMSD scores) with PlUGT15 as compared with models developed with the SWISS-MODEL tool using UGT85H2 as a template.
Together with that, in our recent publication, we have compared protein models developed with Alphafold2 and SWISS-MODEL, and the models developed with Alphafold2 have performed better which has strengthened our trust in Alphafold2 for protein modelling (Citation: Sajid, M.; Stone, S.R.; Kaur, P. Phylogenetic Analysis and Protein Modelling of Isoflavonoid Synthase Highlights Key Catalytic Sites towards Realising New Bioengineering Endeavours. Bioengineering 2022, 9, 609.).
- The differently stained structures in Figure 2 are not clearly explained. And what interactions does the yellow line show? It is recommended to display the key points of all enzymes involved in the paper in the supplementary figures.
Response: The figure (Figure 4; figure number is changed due to the addition of new figures) representing docking results of model UGTs has been improved to show docking orientations of all substrates as well as the interaction of substrates with key amino acids. And, docking results of all UGTs with selected substrates have been added in the supplementary figures.
Captions of Figure 4 improved during the revision process:
Figure 4. Docking results of PlUGT15 (A to F), UGT72V3 (G to L) and UGT73F1 (M to R) with selected substrates. Different orientations of catalytic His in selected models highlights distinct active site structures that result in the different orientation of substrates (the substrates are shown in vertical orientations for better visuals and comparison). Yellow dotted lines represent polar interactions between substrates and amino acids, red dotted lines represent the distance between catalytic His and the closest OH group. The distance is given in â„« units. Key amino acids and substrates are shown in stick models. Catalytic His is coloured blue, Tyr in green and Ser in yellow. The carbon skeleton of biochanin-A, daidzein, formononetin; genistein, liquiritigenin and naringenin is coloured cyan, pink, yellow, salmon, grey, and slate respectively, and the oxygen atoms are coloured red.
Figure S4A to S4C: Docking results of UGTs with selected substrates. Different orientations of catalytic His highlights distinct active site structures that results in different orientation of substrates (the substrates are shown in vertical orientations for better visuals and comparison). Yellow dotted lines represent polar interactions between substrates and amino acids, red dotted lines represent the distance between catalytic His and the closest OH group. The distance is given in â„« units. Key amino acids and substrates are shown in stick models. The carbon skeleton of biochanin-A, daidzein, formononetin; genistein, liquiritigenin and naringenin is coloured cyan, pink, yellow, salmon, grey, and slate respectively, and the oxygen atoms are coloured red.
- The format of the Figure S2 caption needs to be unified with the previous ones.
Response: The figure caption has been formatted as required.
- What is the alignment standard for the models in Figure S3?
Response: Figure S5 (figure number is changed due to the addition of new figures) has been improved. Both PlUGT57 and GmUGT4 are aligned with PlUGT15 (model UGT) to compare and highlight the structural difference that likely controls access of substrates to the acceptor active site.
Captions of Figure S5 have been improved during the revision process:
Figure S5. Structural comparison of PlUGT15 (model UGT, deep salmon colour), PlUGT57 (marine colour) and GmUGT4 (pale green colour). The extended loop regions (shown by the arrowhead) do not let substrates move deep in the acceptor binding cavity (active site), thus the substrate either stays on the surface or accesses the active site through different channels. Protein models of PlUGT57 and GmUGT4 were aligned with PlUGT15 (model UGT) due to the similarity between the models (based on RMSD values) in PyMOL.
- Biochemical experiments to verify the impact of key sites found by MSA and molecular docking on enzyme activity are crucial and necessary.
Response: Yes, as stated/added in the manuscript (lines: 130-132 & lines 155-158) the biochemical experiments are needed to verify the function of key amino acid sites and how they impact OH site specificity and substrate selection. However, due to time limitations, the present study is designed as an in-silico analysis only.
Comments on the Quality of English Language
The logic of the manuscript is not rigorous enough and sometimes not smooth to read.
- The PSPG motif was first introduced in line 79 of the main text, but the introduction of the UGT structural domain is in line 85. And it is necessary to visually display the structural motifs of the N-terminus, C-terminus, etc.
Response: Lines 91-97 have been rewritten to improve the flow of information. And a new supplementary figure has been added to represent N and C terminal domains.
Text from lines 91-97 is given below:
Multiple sequence alignment (MSA) has been performed and results are given in Table 1, Figure 2 and Figure S1. UGT protein structure was divided into two domains: the N-terminal domain, which contains an active site that selects and holds sugar acceptors; the C-terminal domain, which contains plant secondary product glycosyltransferase (PSPG) motif and is responsible for binding with the sugar donor (Figure S2) [22]. Interestingly, the PSPG motif amino acid sequence of Pueraria montana var. lobata UGT1 and PlUGT15 were identical (Figure S3).
Captions of Figure S2 developed during the revision process:
Figure S2. Protein model of PLUGT15 (developed with Alphafold2). The model has been shown as a carton. The N-terminal is coloured pale yellow, the C-terminal is pale green and PSPG motif is coloured marine. UDP-glucose, substrates and catalytic His have been shown in the stick model. UDP-glucose is coloured in salmon and O atoms are shown in red colour. Catalytic His is coloured in rainbow colours. The carbon skeleton of biochanin-A, daidzein, formononetin; genistein, liquiritigenin and naringenin is coloured cyan, pink, yellow, salmon, grey, and slate respectively, and the oxygen atoms are coloured red.
- The 'However' in line 49 of the main text doesn't seem to be quite appropriate logically. The progress made in the synthesis of glycosylated isoflavones in the previous sentence and the low drop rate problem of isoflavone synthesis in this sentence, as well as the contradiction in logic that glycosylation has increased the drop rate of isoflavones.
Response: The word ‘However’ has been removed and the paragraph (lines 60-71, line numbers are changed due to the addition of text during the revision process) has been rewritten to improve clarity.
Text from lines 60-71 is given below:
Recently, focus has shifted to the heterologous biosynthesis of plant natural products, and significant progress has been made in the biosynthesis of isoflavonoids [12–16]. The heterologous biosynthesis of isoflavonoids faces issues of low titer and low yield. Following the development of a microbial strain capable of de-novo biosynthesis or biotransformation of an isoflavonoid of interest, efforts, i.e., directed evolution, and engineering of primary metabolism, are put forward to improve the final yield [12,17]. It has been reported that glycosylation of the end product improves its heterologous biosynthesis rate; therefore, the addition of a UGT gene in the pathway to glycosylate isoflavonoid of interest is a unique approach to enhance the final yield [18]. This is so because isoflavonoids have low solubility and are challenging to transport across the cell/organelles; glycosylation increases the solubility of isoflavonoids, helps in storage and transportation within and out of the cell and drives the biosynthesis pathway forward.
- The introduction from lines 67-72 should include a summary of the research methods used later.
Response: A summary of the research method has been added in lines 79-86 (line numbers are changed due to the addition of text during the revision process).
Text from lines 79-86 is given below:
Thus, the present study is planned to rank selected UGTs for OH site specificity and catalysis efficiency based on multiple sequence alignment (MSA) and protein docking analysis towards O-glycosylation of selected isoflavonoids: genistein (GEN), biochanin-A (BCN), daidzein (DZN), formononetin (FMN) and their parent flavonoids: naringenin (NGN) and liquiritigenin (LQN). The present study will help direct future heterologous biosynthesis of glycosylated isoflavonoids and protein engineering efforts to improve the substrate and site specificity of UGTs.
Reviewer 2 Report
Comments and Suggestions for Authors
The present work is an attempt to analyze the literature data on uridine diphosphate glycosyltransferases (UGTs) with the registered activity of O-glycosylation of isoflavanoids and to rank them according to the specificity of the OH site and the efficiency of catalysis. The article is of interest to specialists, and in the future, the obtained data can be used in protein bioengineering to improve the substrate and site-specificity of UGTs.
Major concern
Introduction. The authors should provide a drawing of the chemical structures of flavonoids and isoflavonoids with the numbering of atoms in the scaffolds. Also, the structures of the compounds selected for the study should be given - isoflavonoids ( genistein (GEN), biochanin-A (BCN), daidzein (DZN), formononetin (FMN)) and their parent flavonoids (naringenin (NGN) and liquiritigenin (LQN)) .
This addition will make the goals and objectives of the article more clear, and will also attract the attention of scientists working in related fields.
I draw your attention to the fact that the article has been sent to the special issue "Chalcones: Biosynthesis, Functions and Biological Significance", therefore, the text of the article requires a note about the biogenetic relationship of chalcones and (iso)flavonoids.
Also, I would recommend the authors to check the English language.
Comments on the Quality of English LanguageAlso, I would recommend the authors to check the English language.
Author Response
Reviewer 2:
Comments and Suggestions for Authors
The present work is an attempt to analyze the literature data on uridine diphosphate glycosyltransferases (UGTs) with the registered activity of O-glycosylation of isoflavonoids and to rank them according to the specificity of the OH site and the efficiency of catalysis. The article is of interest to specialists, and in the future, the obtained data can be used in protein bioengineering to improve the substrate and site-specificity of UGTs.
Major concern
- The authors should provide a drawing of the chemical structures of flavonoids and isoflavonoids with the numbering of atoms in the scaffolds. Also, the structures of the compounds selected for the study should be given - isoflavonoids (genistein (GEN), biochanin-A (BCN), daidzein (DZN), formononetin (FMN)) and their parent flavonoids (naringenin (NGN) and liquiritigenin (LQN)). This addition will make the goals and objectives of the article more clear and will also attract the attention of scientists working in related fields.
Response: A new figure has been added to the main text and the figure caption is given below.
Figure 1. Biosynthetic pathway of isoflavonoids and chemical structures of selected substrates. (A) Chalcone synthase (CHS) converts p-coumaroyl Co-A into naringenin chalcone which is isomerized by chalcone isomerase (CHI) to give flavonoid structure (C6-C3-C6 backbone). Isoflavonoid synthase (IFS) and cytochrome P450 reductase (CPR) perform an aryl ring migration reaction (B-ring) to produce an isoflavonoid backbone from flavonoids. (B) Chemical structures of naringenin (NGN), genistein (GEN), biochanin-A (BCN), liquiritigenin (LQN), daidzein (DZN) and formononetin (FMN) are given. Glycosylation sites have been shown with the letter ‘G’ inside a blue (most common glycosylation sites) or a black circle (less common glycosylation sites).
- I draw your attention to the fact that the article has been sent to the special issue "Chalcones: Biosynthesis, Functions and Biological Significance", therefore, the text of the article requires a note about the biogenetic relationship of chalcones and (iso)flavonoids.
Response: Biogenetic relationship of chalcones and (iso)flavonoids has been added in paragraph 2 (lines: 38 to 42). Additionally, figure 1 (Figure 1. Biosynthetic pathway of isoflavonoids and chemical structures of selected substrates) that has been developed during revision process that also represents the relationship of isoflavonoids to chalcones.
Text from lines 38-42 is given below:
Over 2400 isoflavonoids have been identified, mainly from legumes [6]. Isoflavonoids backbone (C6-C3-C6) is biogenetically derived from flavonoids with the help of isoflavonoid synthase aided by cytochrome P450 reductase. The flavonoid backbone is formed from p-coumaroyl CoA and naringenin chalcone with the help of chalcone synthase and chalcone isomerase, respectively (Figure 1) [6,7].
- Also, I would recommend the authors to check the English language.
Response: English language of the main text has been improved during the revision process.
- Comments on the Quality of English Language
Also, I would recommend the authors to check the English language.
Response: English language of the main text has been improved during the revision process.
Round 2
Reviewer 1 Report
Comments and Suggestions for Authors
The only point I think needs particular attention is a follow-up on question 8. Generally, Alphafold doesn't perform well in relatively flexible regions, which often are catalytic sites. It would be helpful to display the predicted model colored by plddt scores, with a focus on the local confidence of key active motifs. Therefore, the local RMSD of the key motifs would also be valuable.
Comments on the Quality of English Languageline 119: 'Hight of the letters'
Author Response
Reviewer 1:
Comments and Suggestions for Authors
- The only point I think needs particular attention is a follow-up on question 8. Generally, Alphafold doesn't perform well in relatively flexible regions, which often are catalytic sites. It would be helpful to display the predicted model coloured by plddt scores, with a focus on the local confidence of key active motifs. Therefore, the local RMSD of the key motifs would also be valuable.
Response: Thank you so much for the follow-up suggestion. As recommended, a new supplementary figure (Figure S4; figure caption given below) has been developed to represent protein models of selected UGTs coloured as per the pLDDT score and relevant information has been added in the main test (lines 142-419, text given below). Overall, protein models of selected UGTs, key catalytic residues (histidine and aspartic acid), PSPG motif region and donor binding site of all UGT models have shown very high confidence as per the pLDDT score.
The caption of Figure S4 is given below:
- Figure S4. Protein models of selected UGTs developed with Alphafold. The protein models are shown as cartons and coloured as per pLDDT score: blue: >90 score (model confidence very high), cyan: >90 and <70 (model confidence confident), yellow: 70> and <50 (model confidence low), and orange: <50 score (model confidence very low). Catalytic amino acids (histidine and aspartic acid) and PSPG motif region are highlighted with green shade and their pLDDT score has been shown.
The text added in the lines 142 – 149 is given below:
- A very high per-residue confidence score (pLDDT score >90) has been seen for key catalytic residues (Asp), PSPG motif region and donor binding site of all UGT models. For catalytic His, 15 models have shown very high confidence (pLDDT >90) and 7 models have shown good confidence (pLDDT 90> and <70) in His`s predicted orientation and position (Figure S4). Based on the pLDDT score, the predicted models of selected UGT, position and orientation of catalytic residues, PSPG motif region and donor binding site were considered reliable [28].
Ref: Tunyasuvunakool, K.; Adler, J.; Wu, Z.; Green, T.; Zielinski, M.; Žídek, A.; Bridgland, A.; Cowie, A.; Meyer, C.; Laydon, A. Highly Accurate Protein Structure Prediction for the Human Proteome. Nature 2021, 1–9.
- Comments on the Quality of English Language “Line 119: 'Hight of the letters”
Response: The typo has been fixed and the text in line 119 is revised for better clarity.
The revised text in the line 119 is given below:
- Amino acids are shown with single-letter symbols, and the height of the letter at each position represents the degree of conservation.